# RePoseDM: Recurrent Pose Alignment and Gradient Guidance for Pose Guided Image Synthesis

Anant Khandelwal
Glance AI
anant.iitd.2085@gmail.com

## Abstract

*Pose-guided person image synthesis task requires re-rendering a reference image, which should have a photorealistic appearance and flawless pose transfer. Since person images are highly structured, existing approaches require dense connections for complex deformations and occlusions because these are generally handled through multilevel warping and masking in latent space. The feature maps generated by convolutional neural networks do not have equivariance, and hence multi-level warping is required to perform pose alignment. Inspired by the ability of the diffusion model to generate photorealistic images from the given conditional guidance, we propose recurrent pose alignment to provide pose-aligned texture features as conditional guidance. Due to the leakage of the source pose in conditional guidance, we propose gradient guidance from pose interaction fields, which output the distance from the valid pose manifold given a predicted pose as input. This helps in learning plausible pose transfer trajectories that result in photorealism and undistorted texture details. Extensive results on two large-scale benchmarks and a user study demonstrate the ability of our proposed approach to generate photorealistic pose transfer under challenging scenarios. Additionally, we demonstrate the efficiency of gradient guidance in pose-guided image generation on the HumanArt dataset with fine-tuned stable diffusion.*

## 1. Introduction

Skeleton-guided image synthesis presents a formidable challenge within the domain of computer vision, offering a diverse array of applications spanning e-commerce, virtual reality, the metaverse, and the entertainment industries, aimed at content generation. Furthermore, these algorithms hold promise in enhancing performance via data augmentation for downstream tasks such as person re-identification [45]. The main challenge here is to generate photorealistic images that adhere to the specified target pose and appearance derived from the source image. In the literature, a variety of techniques have been introduced, such as generative adversarial networks (GANs), diffusion models (DM), and variational autoencoders (VAE). Generally, all of them used CNNs (convolutional neural networks), which suffer from the problem of spatial transformation [35] and generation of equivariant feature maps [4]. To address the issue of spatial feature transformation, stacked CNNs have been introduced to expand the receptive field from local to global. However, transitioning from local to global can lead to the loss of fine-grained details in the source appearance. Flow-based networks [20, 27, 32] handle efficient spatial transformations, but they exhibit diminished performance in scenarios involving complex deformations and severe occlusions [28]. Since human images are highly structured, addressing complex deformations and occlusions necessitates multi-level warping and masking in latent space. However, this stacked warping approach was still unable to fully resolve the issue, as it tended to diminish the fine-grained details of the source appearance [3, 6]. GAN or VAE methods, whether flow-based or stacked CNNs, encounter challenges related to appearance deformations such as blurry details or low-quality outputs. Conversely, diffusion-based models rely on the iterative reduction of noise, modeled through CNN and attention mechanisms. Attention mechanisms, as described by Vaswani et al. [35], possess the capability of capturing dependencies by directly computing the interactions between any two positions. Therefore, diffusion models should theoretically be able to achieve perfect alignment of the source image and target pose through cross-attention. Unfortunately, in practice, due to the lack of equivariance between feature maps obtained from CNNs, the interaction positions cannot be decoded well by the attention mechanism. Hence, to address this issue, we proposed *Recurrent Pose Alignment* to repeatedly align the source image with a given target pose (shown in Supp. Figure 8). Injecting these pose-aligned features into the cross-attention in U-Net reduce the pose alignment error. It is based on multi-level warping, but it fails in cases of severe occlusions. Therefore, we propose *gradient-guided* training/sampling from diffusion models to further reduce the pose error in a lo-

calized manner. Our major contributions are as follows:

- We proposed a method called *Recurrent Pose Alignment* as a conditional block within the error prediction module in the diffusion model, aiming to reduce the leakage of the source pose into the denoising pipeline. Additionally, we establish the practical groundwork for multi-level warping, necessary for addressing equivariance in CNN feature maps.
- We proposed a novel *Gradient Guidance* technique to enforce the poses generated through interactions with source appearances to adhere to valid pose manifolds.
- We demonstrated improvements in both pose correction and the accurate generation of source appearance in pose-guided person image synthesis on the DeepFashion, HumanArt, and Market-1501 datasets.

## 2. Related Work

With the significant success of diffusion-based models for conditional image synthesis [15], an attempt known as PIDM [2] has been proposed for photorealistic image synthesis given the target pose and source image. PIDM [2] proposed to denoise the noise concatenated with the target pose image, conditioned on the source image. This process aims to inject the source appearance features into the cross-attention of U-Net. However, the feature map produced by CNNs lacks equivariance, causing the attention mechanism to struggle in accurately mapping source features to the target pose. Additionally, some GAN-based methods [10] simply concatenate the source pose, target image, and target pose as input to obtain the target image, leading to feature misalignment. To address this issue, the method proposed in [11] suggests disentangling the guidance from pose and source appearance using skip-connections in U-Net. Another method proposed in [32] introduces deformable skip connections for efficient spatial transformation. This approach breaks down the overall transformation into a series of local affine transformations, thereby addressing the issue of equivariance in CNN feature maps. Flow-based methods [18, 20, 27] are based on the idea of warping the source appearance with the target pose. GFLA [27] proposes global flow fields and an occlusion mask for mapping the source image to the target pose. ADGAN [25] utilizes a texture encoder to extract appearance features for different body parts and feeds them into residual AdaIN blocks to synthesize the target image. PISE [39], SPGnet [24], and CASD [44] all employ parsing maps with variations in encoder and decoder to generate the final image. CoCosNet [43] uses an attention mechanism to extract appearance features between cross-domain images. NTED [29], a method based on distribution operation, proposes obtaining semantically similar texture features by aligning the semantic textures of the source image with the semantic filters for the target pose. All the aforementioned approaches propose a method to map the source to the target pose in some way but do not involve pose correction. We proposed *gradient guidance* to iteratively correct the pose.

## 3. Proposed Method

**Overall Framework**: Fig.1 displays the overall architecture of our proposed generative model, *RePoseDM*. Given a source image $I_S$ and a target pose $P$, our goal is to generate a target image that strictly follows the target pose and has the same appearance as the source. This is achieved using the conditional guidance into U-Net from pose-aligned texture features (*Recurrent Pose Alignment*) and gradient guidance (from *Pose Interaction Fields*) to learn the plausible trajectories for the target pose generation. We will discuss next (Section 3.1) the training of conditional diffusion models with gradient guidance, followed by *Recurrent Pose Alignment* in Section 3.2 and *Pose Interaction Fields* in Sec.3.3.

### 3.1. Appearance Conditioned Diffusion Model

*RePoseDM* is based on the generative scheme of iterative noise reduction as proposed in Denoising diffusion probabilistic model (DDPM) [15]. The general idea behind DDPM is the combination of two processes: *forward* diffusion and *backward* denoising. During the forward (diffusion) process, it gradually adds noise to the data sampled from the target distribution $\boldsymbol{y}_0 \sim \boldsymbol{q}(\boldsymbol{y}_0)$, and the backward (denoising) process aims to eliminate this noise. The forward process is described by the Markov chain with the following distribution:

$$q(\boldsymbol{y}_T|\boldsymbol{y}_0) = \prod_{t=1}^{T} q(\boldsymbol{y}_t|\boldsymbol{y}_{t-1}), \text{ where}$$
$$q(\boldsymbol{y}_t|\boldsymbol{y}_{t-1}) = \mathcal{N}(\boldsymbol{y}_t; \sqrt{1-\beta_t}\boldsymbol{y}_{t-1}, \beta_t\mathbf{I}). \quad (1)$$

Here, the noise schedule $\beta_t$ is an increasing sequence of $t \in [0, T]$ with $\beta_t \in (0, 1)$. Using the notations $\bar{\alpha}_t = \prod_{t=1}^{T} \alpha_t, \alpha_t = 1 - \beta_t$. we can sample from $q(\boldsymbol{y}_t|\boldsymbol{y}_0)$ in a closed form at an arbitrary time step $t$: $\boldsymbol{y}_t = \sqrt{\bar{\alpha}_t}\boldsymbol{y}_0 + \sqrt{1-\bar{\alpha}_t}\epsilon$, where $\epsilon \in \mathcal{N}(0, \mathbf{I})$. For the denoising process, we predict the noise $\epsilon_\theta(\boldsymbol{y}_t, t, \boldsymbol{P}, \boldsymbol{I}_S)$ using the neural network (U-Net) using which we can predict the next-step sample $\boldsymbol{y}_{t-1}$ from the noisy sample $y_t$ at previous time-step given as:

$$\boldsymbol{y}_{t-1} = \frac{1}{\sqrt{\alpha_t}}\left(\boldsymbol{y}_t - \frac{1-\alpha_t}{\sqrt{1-\bar{\alpha}_t}}\epsilon_\theta(\boldsymbol{y}_t, t, \boldsymbol{P}, \boldsymbol{I}_S)\right) + \sigma_t\epsilon \quad (2)$$

where $\epsilon \sim \mathcal{N}(0, \mathbf{I})$, $\sigma_t^2$ is variance which is $\beta$ in DDPM[15]. With this iterative process we sample the clean image $\boldsymbol{y}_0$ from the noise $\boldsymbol{y}_T \in \mathcal{N}(0, \mathbf{I})$.

**Noise Prediction** $\epsilon_\theta(\boldsymbol{y}_t, t, \boldsymbol{P}, \boldsymbol{I}_S)$: Given the target pose $\boldsymbol{P}$

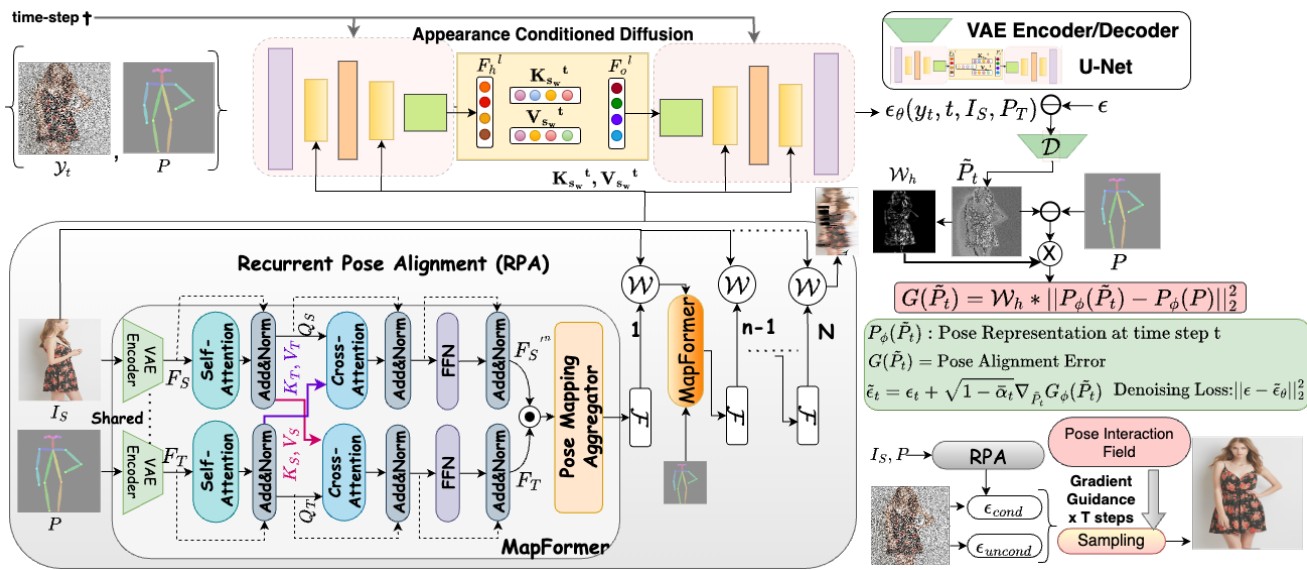

Figure 1. **RePoseDM**: Architecture of our proposed U-Net based Diffusion Model with *Recurrent Pose Alignment* and *Gradient Guidance* from *Pose Interaction Fields*. Warped source appearance features are fed to U-Net using cross-attention.

and noisy sample $y_t$, we concatenate the representation of these two (obtained from VAE encoder) before passing it to the noise predictor network **U-Net**. The target pose $P$ will guide each intermediate denoising step to synthesize the final person image for the given pose and bind the garment texture on that pose using the warped texture features obtained from the *Recurrent Pose Alignment Block* (explained in Section 3.2). This block, recurrently at each step, warps the source image with the target pose $P$ to produce the texture features which can be superimposed onto the target pose in the denoising process. This ensures no source pose leakage and hence avoids any additional noise that can cause pose misalignment. However, as shown in Fig.1, for the challenging transfer of poses i.e. left side pose (source) to the front pose (target), there are deformations in predicted $\tilde{P}_t$ which is termed as *pose alignment noise*. Due to this additional pose alignment noise, we train the denoising process using gradient guidance from *Pose Interaction Fields* (explained in Section 3.3) $G(\tilde{P}_t)$, where $\tilde{P}_t$ is the pose obtained from the denoised image which is obtained after subtracting ground truth noise $\epsilon$ from noise predicted by U-Net i.e. $\epsilon_\theta$. Let $\tilde{\epsilon}_\theta$ be the updated predicted noise including the pose alignment noise, incorporating this in the denoising training loss as:

$$L_{mse} = \mathbb{E}_{t\sim[0,T],y_0\sim q(y_0),\epsilon}||\epsilon - \tilde{\epsilon}_\theta(y_t,t,P,I_S)||^2 \quad (3)$$

where $y_t \sim q(y_t|y_0)$ is the noisy sample from the forward process. For an effective learning strategy in fewer steps, an additional loss term $L_{vib}$ is proposed in [26], to learn the noise variance $\Sigma_\theta$ of the posterior distribution $q(y_{t-1}|y_t) = p_\theta(y_{t-1}|y_t) = \mathcal{N}(y_t; \mu_\theta(y_t,t,P,I_S), \Sigma_\theta(y_t,t,P,I_S))$.

Instead of directly deriving $\mu_\theta, \Sigma_\theta$ from the neural network, we predict only the noise $\epsilon_\theta$ from which the mean and variance are calculated [26]. The updated loss is given as:

$$L_{total} = L_{mse} + L_{vib} \quad (4)$$

## 3.2. Recurrent Pose Alignment

**MapFormer**: Given the source image $I_S$ and the target pose image $P$, these are mapped to their respective latent representations using VAE encoder. These latent representations are then fed to two key components: Attention and Pose Mapping Aggregator. Attention in *MapFormer* consists of both Self-Attention and Cross-Attention blocks for $I_S$ and $P$ respectively. Both blocks perform region-specific attention to introduce the effect of multi-level warping from global $\rightarrow$ non-local region $\rightarrow$ local region. This is inspired by [8, 9, 12, 21], where multi-level warping of the source image is performed to cope with warping deformations due to non-equivariant feature maps for both images. Specifically, for the coordinate point $\mathbf{x}$, the region $\mathcal{N}(\mathbf{x})$, query $Q$, key $K$, and value $V$, the self-attention (self(.)) and cross-attention (cross(.)) are given as follows:

$$\text{self}(\mathbf{x}) = softmax\left(\frac{Q_i(\mathcal{N}(\mathbf{x}))^T K_i(\mathcal{N}(\mathbf{x}))}{\sqrt{D}}\right) V_i(\mathcal{N}(\mathbf{x}))$$

$$\text{cross}(\mathbf{x}) = softmax\left(\frac{Q_i(\mathcal{N}(\mathbf{x}))^T K_j(\mathcal{N}(\mathbf{x}))}{\sqrt{D}}\right) V_j(\mathcal{N}(\mathbf{x}))$$

$$(5)$$

where $i$ and $j$ refer to the indices of images, denoting the query from one and the key, value from another. $D$ is the

depth of the convolution layer calculating the representation of the region $\mathbb{N}(x)$. Similar to previous works [31, 38], residual connection, layer normalization, and feed-forward layers have been used, as shown in Fig.1. Only 1 layer of self-attention and cross-attention is used with different kernel sizes for a fixed depth $D$ of the convolution layer.

**Pose Mapping Aggregator**: Pose aggregation is performed with different kernels to accumulate the interaction between the source image and the target pose from global to localized regions. After calculating the intra/inter correspondences between the source image and the target pose for a given region $\mathbb{N}(\mathbf{x})$, the correlation between localized regions is determined as:

$$\mathbf{C}(\mathbf{x}, \mathbf{r}) = \text{ReLU}(\mathbf{F}'_T(\mathbf{x})^{\mathrm{T}} \mathbf{F}'^n_S(\mathbf{x}+\mathbf{r})) \left\| \mathbf{r} \right\|_\infty \leq R \quad (6)$$

where $\mathbf{F}'^n_S$ is the feature at step $n$ after FFN, residual connection and layer norm as shown in Fig.1. It is obtained from the warped input feature $\mathbf{F}^{n-1}_S$ at previous step $n-1$ (equation 9). $R$ controls the radius of each local region and hence produce the correlation feature of size $H \times W \times (2R+1) \times (2R+1)$. The architecture of pose aggregator is similar to the previous works [3, 4, 31], it consists of 2 convolution layers and 1 max-pooling layer, with convolution layer of same depth $D$ used during attention. Correlation is similar to cross-attention[31] where the output is the mapping of source features to pose-aligned features. At iteration $n$, the output of pose aggregator for specific region is denoted as $\Delta F^n$.

**Recurrent Pose Alignment**: For each region $\mathbb{N}(x)$, the MapFormer and Pose-Aggregator produce the correlation map of pose-specific features for global to local regions in an iterative manner for $n \in [1, N]$. We introduce a recurrent accumulator $\mathcal{F}$ to combine these features for all regions, which can be used to perform warping recursively. This ensures the accurate estimation of warped source features (finally $F_S{}^N$ fed to the U-Net's Cross-Attention as shown in Fig. 1), reducing pose misalignment to the minimum possible. The recurrent accumulation is defined as:

$$\mathcal{F}^{n+1} = \mathcal{F}^n \Delta F^{n+1}, \text{ where} \quad (7)$$
$$\Delta F^{n+1} = \mathcal{M}(F_T, F_S{}^n) \text{ and} \quad (8)$$
$$F_S{}^n = \text{Encoder}(\mathcal{W}(I_S; \mathcal{F}^n)) \quad (9)$$

where $\mathcal{M}$ is MapFormer with Pose Aggregator. For $n = 0, \mathcal{F}^0 = \Delta F^0 = \mathcal{M}(F_T, F_S)$, where $F_S = \text{Encoder}(I_S)$, Encoder is obtained from VAE encoder as discussed above.

### 3.3. Pose Interaction Fields

The warped image generated by recurrent pose alignment still has pose misalignment (as shown in Figure 1), which gets injected into U-Net at every time step $t$. Hence, the predicted noise $\epsilon_\theta(\mathbf{y}_t, t, P, I_S)$ contains this pose alignment

noise, which needs to be explicitly addressed in the loss function during the training of the denoising process. To alleviate this issue, we propose to guide the diffusion model with the distance between interaction poses generated from the diffusion model (shown by $\tilde{P}_t$ in Fig.1) and the valid pose. To calculate the distance, we obtain the pose representation from a pre-trained pose estimator model, i.e., HigherHRNet [7]. Inspired by *HumanSD* [17], we obtain the generated pose map $\tilde{P}_t$ by applying the VAE decoder over the subtraction of noise predicted by U-Net from the ground truth noise, given as:

$$\tilde{P}_t = \text{VAE}_{decoder}(\epsilon - \epsilon_\theta(\mathbf{y}_t, t, P, I_S)) \quad (10)$$

We then use bottom-up pose estimator HigherHRNet [7] (pre-trained on MSCOCO [19] and Human-Art[16]) to get pose representation. Using pretrained model, we are able to localize the locations of human joints from the pose map generated by the diffusion model using the heatmap generated $H \in R^{H \times W \times k}$ where $k$ is the number of joints. Summing and thresholding (threshold$= 0.1$) the heat maps across the joints give the heat map mask $H_M$ which served as the weight $W_h$ indicating exactly where the focus is required to reduce the pose alignment noise. Finally, we pass the pose $\tilde{P}_t$ to the backbone of pose estimator to get the latent representation for the pose generated by the diffusion model and the valid pose. The pose alignment error is then calculated as:

$$G_\phi(\tilde{P}_t) = W_h * ||P_\phi(\tilde{P}_t) - P_\phi(P)||^2_2 \quad (11)$$

Where $P_\phi$ represents the backbone of the pose estimator model, which provides the latent representation of the given pose. We multiply it with the weight $W_h$ to get the error only at those locations where the pose is generated by the diffusion model. Inspired by recent work on neural distance fields to learn valid human pose manifolds [34] and robotic grasping manifolds [37], which provide information about how far an arbitrary pose is from a valid pose, in our case, the offset vector $G_\phi$ takes in the pose generated by the diffusion model and outputs how far it is from the target pose $P$. This offset vector $\Delta P = G_\phi(\tilde{P}_t)$ can generate the manifold of valid interaction poses for the pose generated by the diffusion model at step $t$: $\tilde{P}_t + \Delta P$. To ensure that the sample update from $\mathbf{y}_t \rightarrow \mathbf{y}_{t-1}$ lies on the correct pose manifold, we leverage gradient guidance as follows:

$$\mathbf{y}'_{t-1} = \mathbf{y}_{t-1} \underbrace{-\nabla_{\tilde{P}_t} G_\phi(\tilde{P}_t)}_{\Delta y_{0,t}} \quad (12)$$

where the next step sample $\mathbf{y}_{t-1}$ is generated using equation 2. Specifically, if we add $\Delta y_{0,t}$ from equation 19 in equation 2, then following [1] the original forward equation $y_t = \sqrt{\bar{\alpha}_t} y_0 + \sqrt{1 - \bar{\alpha}_t} \epsilon$ is converted to:

$$y_t = \sqrt{\bar{\alpha}_t}(\hat{y}_{0,t} + \Delta y_{0,t}) + \sqrt{1 - \bar{\alpha}_t} \tilde{\epsilon}_t \quad (13)$$

where $\hat{y}_{0,t}$ is the estimated cleaned image using Tweedie's formula[33], perturbed diffusion model given as:

$$\tilde{\epsilon}_t = \epsilon_t - \sqrt{\bar{\alpha}_t/(1-\bar{\alpha}_t)}\Delta y_{0,t}, \qquad (14)$$

where $\Delta y_{0,t} = -\eta \nabla_{\tilde{P}_t} G_\phi(\tilde{P}_t)$ for some scale factor $\eta$, leading to

$$\tilde{\epsilon}_t = \epsilon_t + \sqrt{1-\bar{\alpha}_t}\nabla_{\tilde{P}_t} G_\phi(\tilde{P}_t) \qquad (15)$$

suggesting the pose guided diffusion model. The previous work [1] sets the scale factor to $\eta = \sqrt{1-\bar{\alpha}_t}$.

### 3.4. Sampling and Gradient Guidance

At test time, the samples are generated from the model with the input random noise $\boldsymbol{y}_T = \mathcal{N}(0, \mathbf{I})$ concatenated with the target pose $P$ and the interaction conditioning $F_S^N$, which contains features of the source garment when interacted with the target pose $P$. The distribution $p(y_{t-1}|y_t, P, I_S)$ governs the sampling process from $t = T$ to $t = 0$ in iterative manner. Following, the previous works [14] we also leveraged the classifier-free guidance. This implies that we have conditional and unconditional sample from the model and combine them as follows

$$\epsilon_{cond} = \epsilon_{uncond} + w_c * (\epsilon_{interaction} + \sqrt{1-\bar{\alpha}_t}\epsilon_{gradient}) \qquad (16)$$

where the strength of interaction conditioning is controlled by scalar $w_c$, the unconditioned prediction is given as $\epsilon_{uncond} = \epsilon_\theta(\boldsymbol{y}_t, t, \emptyset, \emptyset)$ where the condition of target pose $P$, and source image $I_S$ has been set to null, which implies there is no interaction conditioning $F_S^N$, on the other hand the conditioned prediction is $\epsilon_\theta(\boldsymbol{y}_t, t, P, I_S)$, where the interaction features $F_S^N$ are given as input to cross-attention of U-Net. Additionally, we leveraged gradient guidance for the pre-trained pose estimation model, to account for pose alignment error during denoising process, this is given as $\epsilon_{gradient} = \nabla_{\tilde{P}_t} G_\phi(\tilde{P}_t)$, which is combined as in eqn 15. During training, the diffusion model learns both conditioned and unconditioned distributions by randomly setting $P, I_S = \emptyset$ for $p\%$ of examples, which learns the distribution $p(\boldsymbol{y}_0)$ faithfully and leverage it during inference.

## 4. Experiments

**Datasets**: Experiments were carried out on the In-Shop Clothes Retrieval Benchmark of DeepFashion [22] and the Market-1501 [42] datasets. The DeepFashion dataset contains $52,712$ high-resolution images of men and women fashion models, while the Market-1501 dataset contains $32,668$ low-resolution images. Pose skeletons were extracted using OpenPose [5]. For DeepFashion, following the dataset splits in [45], there are a total of $101,966$ pairs

| Dataset | Methods | FID($\downarrow$) | SSIM($\uparrow$) | LPIPS($\downarrow$) |
|---|---|---|---|---|
| DeepFashion (256 × 176) | Def-GAN | 18.457 | 0.6786 | 0.2330 |
| | PATN | 20.751 | 0.6709 | 0.2562 |
| | ADGAN | 14.458 | 0.6721 | 0.2283 |
| | PISE | 13.610 | 0.6629 | 0.2059 |
| | GFLA | 10.573 | 0.7074 | 0.2341 |
| | DPTN | 11.387 | 0.7112 | 0.1931 |
| | CASD | 11.373 | 0.7248 | 0.1936 |
| | NTED | 8.6838 | 0.7182 | 0.1752 |
| | PIDM | 6.3671 | 0.7312 | 0.1678 |
| | **RePoseDM (Ours)** | **5.1986** | **0.7923** | **0.1463** |
| DeepFashion (512 × 352) | CocosNet2 | 13.325 | 0.7236 | 0.2265 |
| | NTED | 7.7821 | 0.7376 | 0.1980 |
| | PIDM | 5.8365 | 0.7419 | 0.1768 |
| | **RePoseDM (Ours)** | **4.7284** | **0.7944** | **0.1566** |
| Market-1501 (128 × 64) | Def-GAN | 25.364 | 0.2683 | 0.2994 |
| | PTN | 22.657 | 0.2821 | 0.3196 |
| | GFLA | 19.751 | 0.2883 | 0.2817 |
| | DPTN | 18.995 | 0.2854 | 0.2711 |
| | PIDM | 14.451 | 0.3054 | 0.2415 |
| | **RePoseDM (Ours)** | **13.689** | **0.3284** | **0.2206** |

Table 1. Quantitative Comparision of **RePoseDM** with SOTA methods in terms of FID, SSIM, LPIPS

in the training set and $8,570$ pairs in the testing set. For both datasets, the training and test sets do not contain overlapping person identities. The images from both datasets vary in factors such as illumination, background, and viewpoint angles. Evaluation metrics are explained in App. 7 and implementation details are given in App. 8.

### 4.1. Quantitative and Qualitative Comparisons

We compared our model *RePoseDM* both qualitatively (Fig.2, 3) and quantitatively (Tab.1) with several state-of-the-art methods, namely, PIDM [2], NTED [29], CocosNet2 [43], CASD [44], DPTN [40], Def-GAN [32], PATN [45], ADGAN [25], PISE [39], and GFLA [27]. For qualitative comparison, we experimented with the DeepFashion dataset at multiple resolutions, i.e., $256\times176$ and $512\times352$ images. For Market-1501, we used $128 \times 64$ images. Tab.1 shows that our model achieved the best results in all three metrics: FID, SSIM, and LPIPS, indicating superior photorealism in the generated images (Fig.2), while preserving finer details like texture and cloth warping for the given target pose. Moreover, the improvement over SSIM and LPIPS demonstrates better structural properties, not only in terms of generating the target pose but also in accurate garment reconstruction for a given target pose. Fig.2 presents a qualitative visual comparison of our method with state-of-the-art frameworks on the DeepFashion dataset. The synthesized images from other methods were obtained using the pre-trained models provided by the corresponding authors. From the results, it can be observed that ADGAN [25] and PISE [39] consistently perform poorly in terms of the quality of generated images and cannot retain shape and texture

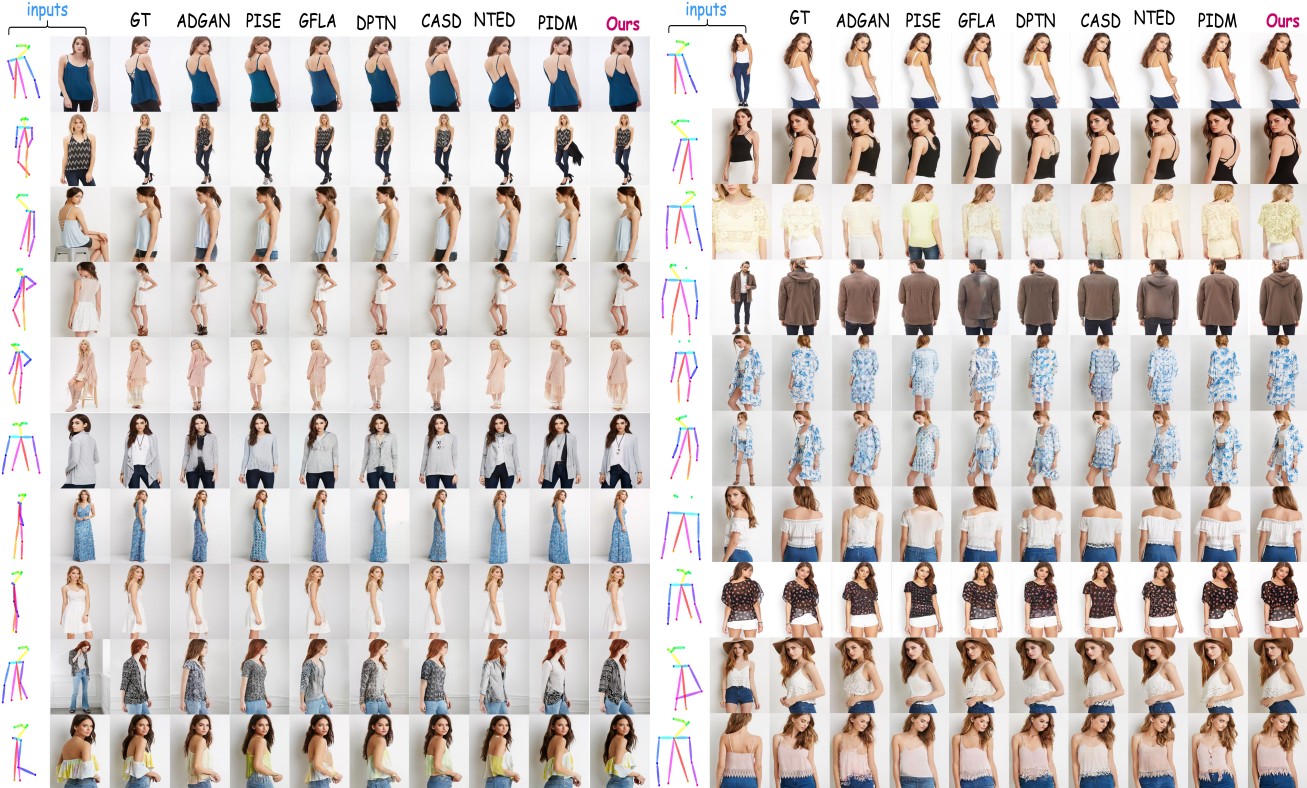

Figure 2. Qualitative comparison of several SOTA methods on the DeepFashion dataset. The inputs shown are target pose and source image, ground truth shows the image in target pose. Images generated from several methods are shown next. Ours indicate RePoseDM

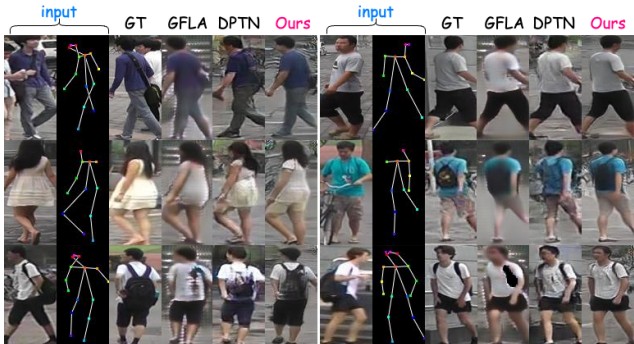

Figure 3. Qualitative comparision from several SOTA methods are shown on Market-1501 dataset. Ours indicate RePoseDM

details. GFLA [27] improves on the texture details (4th and 8th row left columns, and 8th row right column) but still struggles to generate reasonable results in unseen regions of images (6th row left column, and 2nd and 4th row right columns). CASD [44] and NTED [29] show slight improvements compared to methods before them but still struggle to preserve the source appearance in complex scenarios (9th and 1st rows in left column, and 3rd, 5th, and 10th rows in right column). Although PIDM [2] generates sharp images,

it either has problems with incorrect pose (5th and 7th row right columns, 5th row left column) or missing texture (2nd, 4th, 6th, 9th, and 10th row left columns, and 2nd, 3rd, 6th, 8th, and 10th row right columns). Our method minimizes texture errors and faithfully generates images for invisible regions as well. With pose guidance, our method generates the correct pose in those places where PIDM [2] missed, as shown in Fig.2 (5th and 7th row right columns, 5th row left column). Additionally, we provide qualitative results on the Market-1501 dataset as shown in Fig.3. The Market-1501 dataset contains challenging backgrounds, and as shown in Fig.3, our method is still able to generate photorealistic images while faithfully retaining the source appearance in the target pose. Additionally, some results on appearance control are shown in Fig.4, where given the reference image needs to be transformed to wearing a garment provided in the garment image. We chose challenging poses of garment images and demonstrated that our method faithfully generates those invisible regions in a reference image.

## 4.2. Human Perception Study

To validate the effectiveness of our method in terms of human perception, the results are evaluated by 100 participants in two schemes: 1) For comparison with ground truth

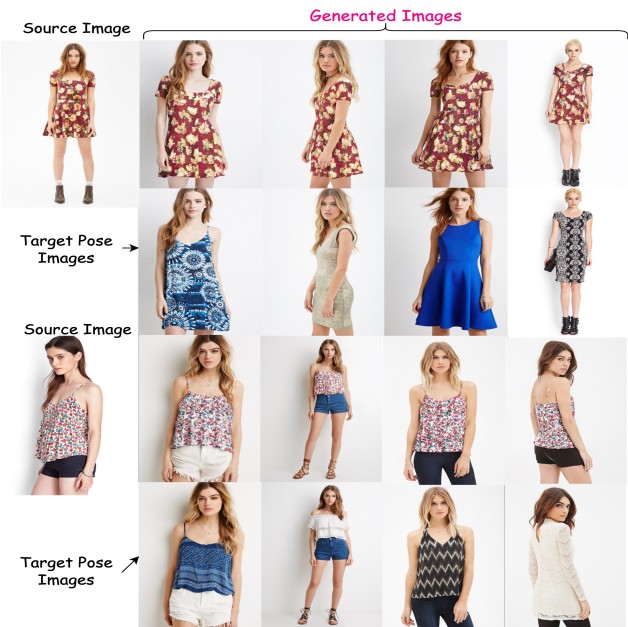

Figure 4. Editing Capability of RePoseDM by controlling garment appearance of target image using source image.

| Methods | FID(↓) | SSIM(↑) | LPIPS(↓) |
|---------|--------|---------|----------|
| B1 | 7.5274 | 0.7189 | 0.1912 |
| B2 | 6.9113 | 0.7055 | 0.1634 |
| RePoseDM | **5.1986** | **0.7923** | **0.1463** |

Table 2. Ablation Studies showing the impact made *Recurrent Pose Alignment* and *Gradient Guidance*

images, we randomly selected 30 images from the test set and 20 images generated by our method. Participants are required to mark the image generated by our method or from ground truth image. Following [2, 44, 45], we adopted two metrics, namely, *R2G* and *G2R*. *R2G* is the percentage of real images classified as generated images, and *G2R* is the percentage of generated images classified as real images. 2) For comparison with other methods, we randomly selected 30 images from the test set and finally presented 30 such sets containing the source image, target pose, image generated by our method, and baselines. Following [2], we quantified this in terms of a metric named *JAB*, which represents the percentage of images from our method that are considered best among all other methods. For each of the three metrics, higher values indicate better performance of our method. Results presented in Fig.7 demonstrate that *RePoseDM* outperforms all other methods. For instance, *RePoseDM* achieves *G2R* = 62%, which is nearly 15% better than the second-best model, and the *JAB* metric is 82%, favoring images generated by our method over others.

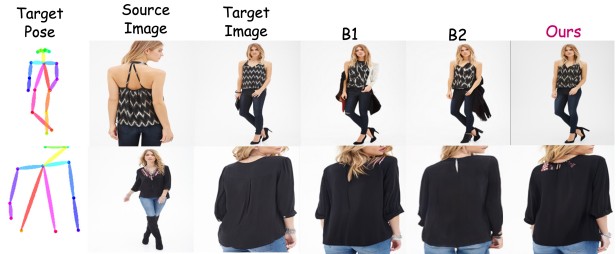

Figure 5. Qualitative comparison with ablation baselines B1 & B2

| Models | Image Quality | Pose Accuracy | | | |
|--------|---------------|---------------|-----|-----|-----|
| | FID↓ | AP↑ | AP(m)↑ | CAP↑ | PCE↓ |
| HumanSD | 26.28 | 31.85 | 24.95 | 59.11 | 1.61 |
| RePoseSD | 24.83 | 33.79 | 26.85 | 61.25 | 1.56 |

Table 3. Quantitative comparison b/w HumanSD and RePoseSD

## 4.3. Ablation Study

We conducted detailed ablation studies to demonstrate the merits of contributions made towards pose-guided person image synthesis. Tab.2 shows the quantitative results on the DeepFashion dataset, comparing *RePoseDM* with several ablation baselines to quantify the contributions made by *Recurrent Pose Alignment* and *Gradient Guidance*. The baseline B1 is the baseline without *Gradient Guidance* and *Recurrent Pose Alignment*. Specifically, it consists of a conditioned U-Net-based noise prediction module where noise and the target pose are concatenated as input, and the source image is passed through a simple CNN encoder to provide texture conditioning to the U-Net. Baseline B1 is PIDM only. Baseline B2 incorporates pose alignment conditions into the U-Net model from the *Recurrent Pose Alignment* block but without any gradient guidance. Finally, RE-POSEDM is compared with two baselines, B1 and B2, as shown in Tab.2 and Fig.5. Qualitatively, as shown in Fig.5, baseline B1 generates images with incorrect pose and texture, while baseline B2 generates images lacking in pose quality.

## 4.4. Effectiveness of Gradient Guidance

We further applied and tested the proposed *Gradient Guidance* in the pre-trained text-based diffusion model, Stable Diffusion v2-1 (SD) [30]. We fine-tuned the SD model on 0.2M text-image-pose pairs obtained from the LAION-Human dataset [17] and tested it on the validation set of the Human-Art dataset [16]. Specifically, we concatenated the noise and pose representation obtained from the VAE encoder and input these to the text-conditioned U-Net for noise prediction at each step of the denoising process. During training, we modified $\epsilon_t$ to $\tilde{\epsilon}_t$ according to Equation 15 containing gradient guidance for predicted pose represen-

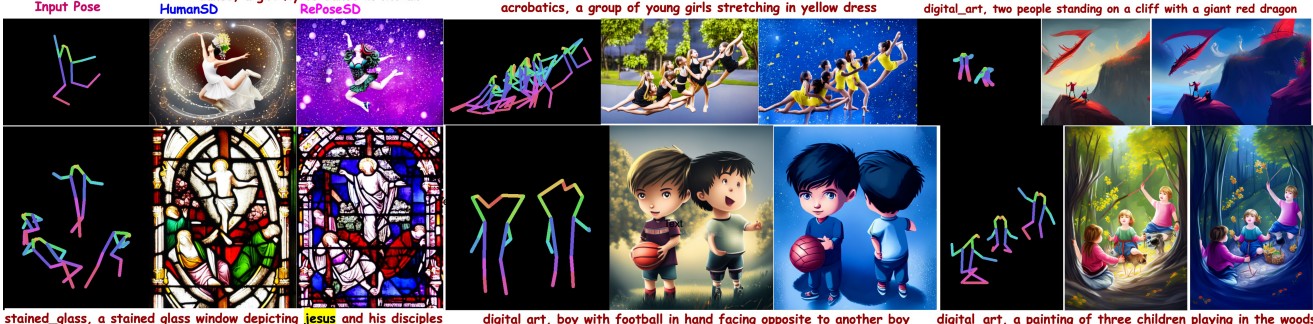

Figure 6. Qualitative comparison b/w HumanSD and RePoseSD. HumanSD and RePoseSD trained on 0.2M text-image-pose pairs randomly sampled from *LAION-Human*

tation obtained from the pre-trained pose estimator HigherHRNet [7]. The pose-guided training loss for stable diffusion is given as:

$$L_{LDM} = E_{t,z,\epsilon} \left[ ||\epsilon - \tilde{\epsilon}_\theta(\sqrt{\bar{\alpha}_t}z_0 + \sqrt{1 - \bar{\alpha}_t}\epsilon, c, t)||_2^2 \right]$$
(17)

where $z_0$ is the latent embedding of a sample $x_0$, $c$ is the text (prompt) embedding, $\bar{\alpha}_t$, $\epsilon_\theta$ and $\epsilon$ is the same as that in vanilla diffusion models. Fig.6 presents the qualitative comparison of pose-guided image generation obtained from *HumanSD* (shown in the middle after the target pose) and REPOSESD (shown in the right after the image from HumanSD). We name our model REPOSESD, which is obtained after fine-tuning with our proposed gradient guidance on the SD model. Clearly, the images generated by our method show better pose alignment, especially in cases where the target pose comprises multiple poses. For example, consider the case of Jesus: our method is able to generate the correct pose of the hands and legs of the other two persons. Similarly, in other cases like dance, where the hand should be close to the head, and football, where the boys should have opposite poses, our method accurately captures these details. Tab.3 shows the quantitative comparison of our method with HumanSD [17], which is the current state-of-the-art in text-based pose-guided generation. Our method outperforms in terms of all three metrics: image quality, pose accuracy, and Text Consistency, using the same metrics as used in [16].

### 4.5. Person Re-Identification

We evaluate the applicability of images generated from *RePoseDM* in improving the performance of the downstream task of person re-identification (re-ID) with data augmentation using images generated from our method. We perform this evaluation with data augmentation on the Market-1501 dataset. Specifically, we randomly sample 20%, 40%, 60%, and 80% from the training dataset and augment each set with images generated from our method for the ran-

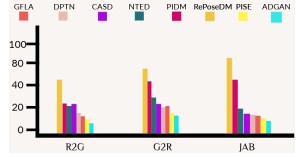

Figure 7. Human Perception Study on DeepFashion dataset showing R2G, G2R and JAB metrics. RePoseDM outperforms other methods.

Table 4. Person re-ID results showing mAP scores of ResNet50 trained on augmented with images generated from several methods

| Models | Real Images(%) | | | |
|---|---|---|---|---|
| | 20% | 40% | 60% | 80% |
| PTN | 54.9 | 56.7 | 66.5 | 71.9 |
| GFLA | 58.1 | 60.1 | 68.2 | 73.4 |
| DPTN | 58.1 | 62.6 | 69.0 | 74.2 |
| PIDM | 61.5 | 64.9 | 71.8 | 75.7 |
| RePoseDM | 63.6 | 65.5 | 72.4 | 76.1 |

domly selected source image and randomly selected target pose. To facilitate the comparison of our method with other baselines, we repeat the above procedure for augmented set creation from images generated with other baselines as well. Specifically, we select PIDM [2], PTN [45], GFLA [27], and DPTN [40]. Tab.4 presents the results using the ResNet50 backbone on each of the individual sets with data augmentation from different methods. The performance of the fine-tuned ResNet50 on data augmented with our method outperforms all the other methods.

### 5. Conclusion

In conclusion, we present a novel approach for pose-guided person image synthesis, leveraging recurrent pose alignment and gradient guidance from pose interaction fields. By providing pose-aligned texture features as conditional guidance, our method achieves photorealistic results with flawless pose transfer, even in challenging scenarios. Extensive experiments on large-scale benchmarks and a user study demonstrate the effectiveness of our approach in generating high-quality images. Furthermore, our method showcases efficiency and stability in generating pose-guided images, as demonstrated on the HumanArt dataset with fine-tuned stable diffusion. Overall, our approach advances the state-of-the-art in pose-guided image synthesis, offering applications in various fields such as virtual try-on, digital entertainment, and augmented reality.

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
