# RePoseDM: Recurrent Pose Alignment and Gradient Guidance for Pose Guided Image Synthesis

## Supplementary Material

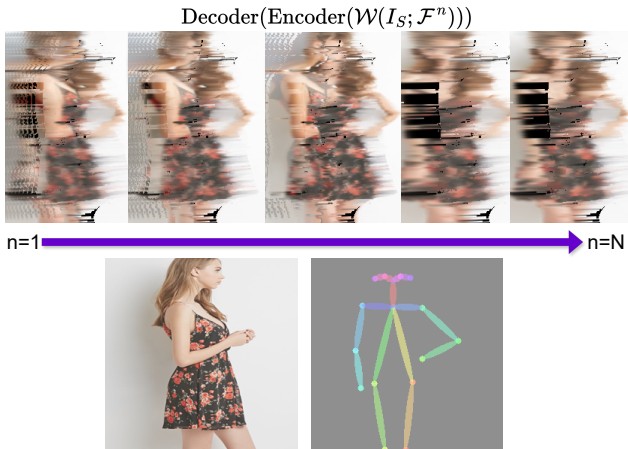

Figure 8. VAE Decoder Output of Recurrent Pose Alignment (RPA) for N iterations, depicting the ability to transfer pose as well as texture

## 6. Motivation

The figure 8 depicts the functioning of the CNN-based Recurrent Pose Alignment. Clearly, the decoded and warped RGB image contains the target pose while retaining the textures of the input image (red color flowers). However, it still contains minor traces of the pose from the input image. We can see the effectiveness of Recurrent Pose Alignment from the decoded output which filters out the source pose as the iterations approaches N. This is due to multi-level warping applied on the source image after recurrent accumulation. The U-Net of the diffusion model is trained solely using the mean squared error between the initial noise and the predicted noise, given as:

$$L_{mse} = \mathbb{E}_{t \sim [0,T], \boldsymbol{y}_0 \sim \boldsymbol{q}(\boldsymbol{y}_0), \epsilon} ||\epsilon - \tilde{\epsilon}_\theta(\boldsymbol{y}_t, t, P, I_S)||^2 \quad (18)$$

Since the above equation for predicted error is in no way related to the pose leakage error, hence the predicted image with source pose error is shown in Figure 5 with Baseline B2. Due to this, we have added the error related to pose generated at each time step and guide the U-Net towards the actual pose. We called the poses generated at each time step the interaction poses. Since the valid interaction pose is already known, we derive the error related to the pose as well. If this error is added at each time step the diffusion learns to minimise the pose error as well. We update the sample generated at each time step with the gradient of the

pose error to drive the changes in the direction of maximum pose error with respect to the target pose. This is depicted in the paper with the equation given as:

$$\boldsymbol{y}'_{t-1} = \boldsymbol{y}_{t-1} \underbrace{-\nabla_{\tilde{P}_t} G_\phi(\tilde{P}_t)}_{\Delta y_{0,t}} \quad (19)$$

Hence, the combined effect of Recurrent Pose Alignment and Gradient Guidance prove the effectiveness in pose guided image generation.

## 7. Evaluation Metrics

Model performance is evaluated using three different evaluation metrics: Structural Similarity Index Measure (SSIM) [36], Learned Perceptual Image Patch Similarity (LPIPS) [41], and Fréchet Inception Distance (FID) [13]. Collectively, SSIM and LPIPS are used to capture reconstruction accuracy. SSIM calculates pixel-level image similarity, while LPIPS computes the perceptual distance between the generated images and reference images by employing a network trained on human judgments. FID calculates the Wasserstein-2 distance between distributions of the generated images and the ground-truth images, it quantifies the realism of the generated images.

## 8. Implementation Details

All experiments were carried out on 8 NVIDIA A100 GPUs. We trained our diffusion model for 300K iterations with a batch size of 8 using the AdamW optimizer [23] with a learning rate of $10^{-4}$. For training with unconditional guidance, we set $p = 10$. We used $T = 1000$ diffusion steps with a linear noise schedule. The diffusion step $t$ was sampled from a uniform distribution at each training iteration. Moreover, during training, we adopted an exponential moving average (EMA) of denoising network weights with a decay rate of 0.9999. For RPA, we use N=5. For sampling, the value of conditional guidance was set to $w_c = 2$. For the DeepFashion dataset, we trained our model using $256 \times 176$ and $512 \times 352$ images. For Market-1501, we used $128 \times 64$ images.

## 9. High Resolution Results

We provide the high resolution results for pose guided image synthesis is shown in Figure 9 and 10.

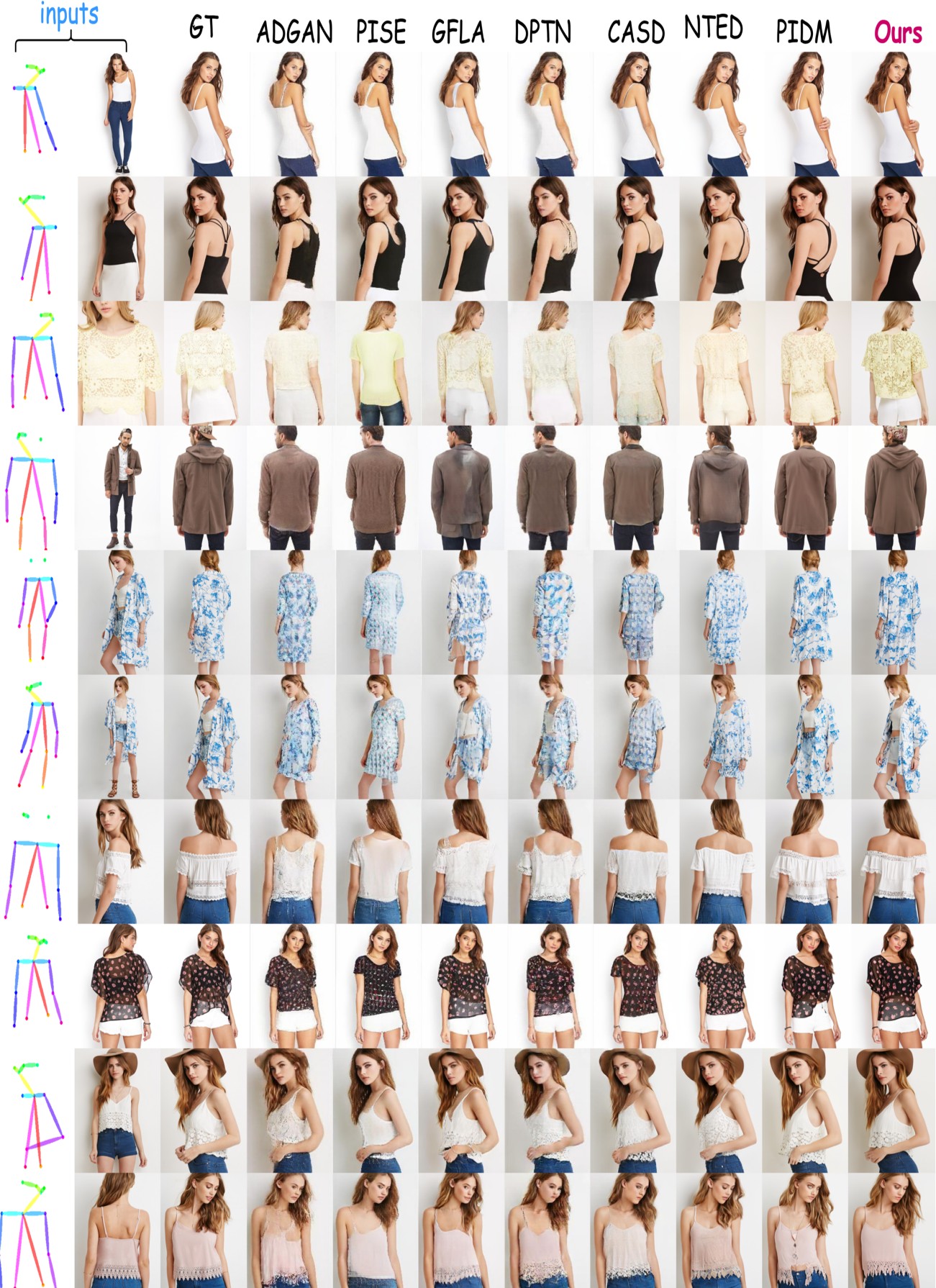

Figure 9. Qualitative comparison of several SOTA methods on the DeepFashion dataset. The inputs shown are target pose and source image, ground truth shows the image in target pose. Images generated from several methods are shown next. Ours indicate RePoseDM

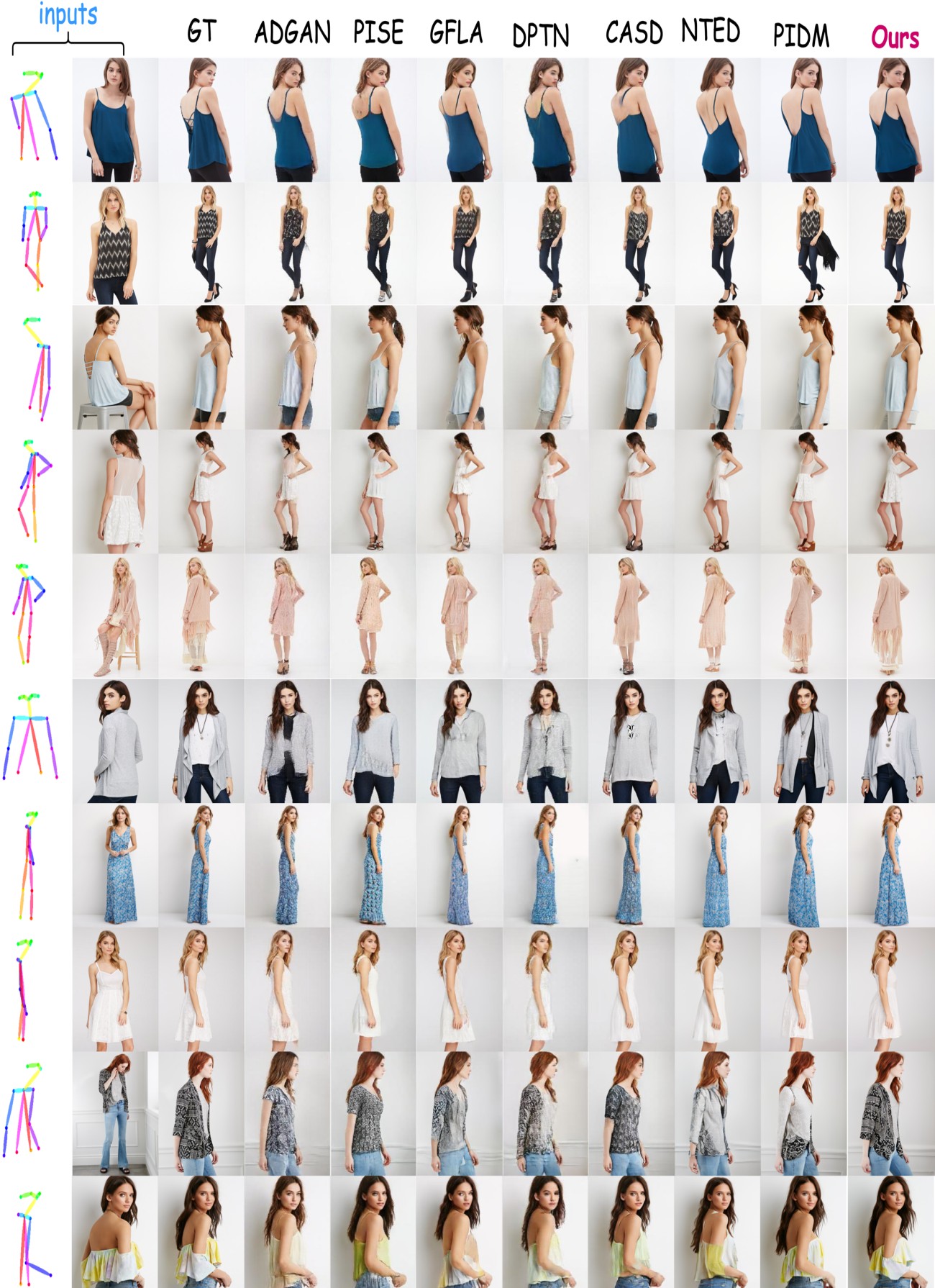

Figure 10. Qualitative comparison of several SOTA methods on the DeepFashion dataset. The inputs shown are target pose and source image, ground truth shows the image in target pose. Images generated from several methods are shown next. Ours indicate RePoseDM