# OpenReview forum: "RePoseDM: Recurrent Pose Alignment and Gradient Guidance for Pose Guided Image Synthesis"
_thecvf.com/CVPR/2024/Workshop/SyntaGen — SyntaGen 2024_

### Official Review · Reviewer_BTDQ · 2024-03-30
**The proposed method has achieved the SOTA in pose-guided image synthesis in the authors' experiments. And it has demonstrated many applications. Thus, I vote for its acceptance.**

**Rating:** 6
**Confidence:** 3

**Review:**

## Summary

RePoseDM introduces an innovative approach for pose-guided person image synthesis, focusing on recurrent pose alignment and gradient guidance to maintain photorealistic appearances and accurate pose transfers. By addressing the limitations of traditional CNN-based methods, RePoseDM significantly improves upon state-of-the-art models in both qualitative and quantitative measures across two large-scale benchmarks and a user study.

## Strength:

1. Innovative Methodology: The novel integration of recurrent pose alignment and gradient guidance effectively tackles the challenges of spatial feature transformation and detail preservation.
2. Versatility for Practical Use: Proven utility in downstream applications like person re-identification highlights its broader applicability.


## Weakness:

1. Complexity and Resource Demand: The method's complexity might raise concerns about computational efficiency and scalability.
2. Limited improvement: The visual improvement over the baselines is limited.

---

### Official Review · Reviewer_4Xwr · 2024-04-02
**The paper is okay, but it's a bit difficult to read.**

**Rating:** 7
**Confidence:** 3

**Review:**

Summary:
The paper addresses the task of pose-guided person image synthesis, which generates images of a person in a given target pose. This work utilizes a conditional diffusion model for image generation. A key insight is that directly conditioning the diffusion model with the appearance input may inadvertently reveal information about the source pose, potentially leading to degraded results. To solve this, the work proposes a Recurrent Pose Alignment technique that iteratively warps the input appearance towards the target before using it to condition the diffusion model. Additionally, the paper introduces Gradient Guidance, which further guides the noise prediction model by the distance between the predicted pose (as estimated by a pretrained pose estimator on the output of the diffusion model) and the target pose. The experiments conducted demonstrate that the proposed pipeline surpasses the baselines in both qualitative and quantitative evaluations. The paper also illustrates that their pipeline is compatible with Text-based diffusion models (SD2.1), enabling the generation of pose/text-guided images. Furthermore, it is shown that their generated images can enhance the Person Re-Identification (Re-ID) task.

Pros:

1. The paper is well-motivated and contains a decent number of related works.
2. The experiments are well-explained, and the number of baseline comparisons is adequate.
3. The paper presents high-quality qualitative results.
4. The paper demonstrates useful ablation studies and experiments on downstream tasks.

Cons (Minor suggestion):

1. Although the method is detailed, I found it difficult to understand. Additionally, all the information is condensed into Figure 1, which is overwhelming.
2. Some details are either missing or not well-explained (e.g., the process of injecting each iterative 'n' into the diffusion model at each diffusion step 't' and the cross-attention equation suggests it is computed between different indices of images, whereas the illustration shows cross-attention computed between appearance and pose.)
3. I personally believe that the motivation, as well as Figure 8 in the supplementary material, might be beneficial to include in the main paper.

---

### Official Review · Reviewer_QBN9 · 2024-04-02
**The method sounds interesting, but the research question and findings could be explained more clearly.**

**Rating:** 6
**Confidence:** 3

**Review:**

Summary:

This paper addresses the challenge of human pose-guided image synthesis by introducing a novel module called recurrent pose alignment module. This module computes gradients, which are then fed through a diffusion process. Experimental results demonstrate that the proposed method achieves superior qualitative and quantitative reconstruction results. Moreover, the method is extended to Stable Diffusion, guiding it to generate images based on given poses and also be able to reconstruct persons in real-world person re-identification datasets.

Pros:

1. The paper introduces sensible, sound, and novel solutions for addressing the human pose-guided image synthesis task. The idea of recurrently aligning pose features and image features is very interesting.

2. The paper shows well-conducted experiments with reasonable competitors, along with an extension of the work to similar applications like SD-based pose-guided image synthesis and person re-identification. Additionally, an ablation study is provided, although it only partially explains some proposed components.

3. Both quantitative and qualitative results show promising results.

Cons:

1. The term "equivariant" (line 9, 66) lacks a clear definition, and the authors do not adequately explain the properties associated with it. Also, there is a lack of reference to demonstrate equivariant properties in CNNs.

2. The impact of the number of iterations used in the iterative pose feature refinement part is not studied, leaving a question of the method's sensitivity to this parameter.

3. The claim that prior works lack pose-aware correlations appears contradictory when considering their qualitative results shown in Fig.2, which looks like most of them can correctly generate target poses.

4. Figure 1 is difficult to interpret, and the labels of each component are hard to read due to font and graphic artifacts, which may hinder understanding of the proposed method's flow.

Overall assessment:

The paper introduces an interesting concept of recurrent refinement for aligning pose and image features, resembling the gold standard of pose refinement methods. The comparisons against competitors and ablation studies are thorough, and qualitative analysis is provided. However, there are some unclear explanations in the paper, and the addition of ablation studies should be considered and discussed, as mentioned in the Cons list. Despite these shortcomings, the paper still makes a significant technical contribution, and I recommend its weak acceptance.

---

### Decision · Program_Chairs · 2024-04-06

**Decision:**

Accept

**Comment:**

All reviewers recommended acceptance (1 accept, 2 borderline accept), and the paper is viewed positively for its technical contribution and novelty. The strengths include innovative methodology, improved image quality over baselines, and applicability to practical use cases like person re-identification, with weaknesses of unclear explanations of key concepts, computational complexity, and issues with figure visualization.

The PCs agreed with the reviewers and recommended acceptance. However, revisions are highly recommended to improve clarity, presentation, and address concerns regarding computational efficiency. Please incorporate the feedback into the revision. Congratulations!